# A Comprehensive Overview of Polypharmacy in Elderly Patients in Saudi Arabia

**DOI:** 10.3390/geriatrics4020036

**Published:** 2019-05-15

**Authors:** Aseel Alsuwaidan, Norah Almedlej, Sawsan Alsabti, Omamah Daftardar, Fawzi Al Deaji, Ali Al Amri, Salem Alsuwaidan

**Affiliations:** 1College of Pharmacy, Princes Nourah University, Riyadh KSA 84428, Saudi Arabia; asee1_@hotmail.com (A.A.); almedlejnorah@gmail.com (N.A.); sawsands.1@gmail.com (S.A.); o-_-@live.com (O.D.); 2Pharmacist Consultant in Prince Sultan Military Medical Center, Riyadh KSA 84428, Saudi Arabia; faldeagi@psmmc.med.sa; 3Family Medicine Consultant, King Abdullah University Hospital, Princes Nourah University, Riyadh KSA 84428, Saudi Arabia; AFAlamri@kaauh.edu.sa; 4Clinical Research Consultant in Health Science Research Center, Princes Nourah University, Riyadh KSA 84428, Saudi Arabia

**Keywords:** geriatrics, elderly patients, polypharmacy, appropriate medication, prescribed medication

## Abstract

**Background/Objectives:** Saudi Arabia has a great percentage of geriatric patients associated with multiple chronic diseases who require close attention and monitoring for their medications. The purpose of this study is to develop a full-framed picture about the utilization of medications for geriatric patients and how to provide better health-care management. **Methodology:** A retrospective cross-sectional study targeting patients 65 years of age and older, who are taking multiple chronic medications for different indications. Descriptive analysis and frequency of the main variables were used as appropriate. Only qualified and professional candidates were chosen for data entry to present the quality and accuracy of data. **Results:** A total of 3009 patient profiles were analyzed, with the patients’ average age in years being 73.26 ± 6.6 (SD). It was found that 55% of the patients have polypharmacy. An average of 6.4 medications were prescribed for patients aged between 65 and 70 years compared with a significant difference for patients aged 71 years and above, while a linear correlation between age and comorbidity diseases associated with all elderly patients. Hypertension, hyperlipidemia, and diabetes mellitus are the most common comorbidity diseases for elderly patients aged 65 years and older. **Conclusion:** Polypharmacy in geriatrics is defined as a patient aged 65 years and older receiving five or more appropriate medications. It is the responsibility of health-care professionals to reduce the number of medications in elderly patients. Awareness of geriatric medications and diagnosed diseases will improve managing adverse drug reaction and other risk factors. Awareness of geriatric medications should elaborate on how to avoid adverse drug reaction and other risk factors. It is the responsibility of physicians and pharmacists to reduce the number of medications in elderly patients. We also prove that the number of medications will not necessarily increase with age. The main impact of this study is to follow the main recommendations to improve health care management in geriatrics.

## 1. Introduction

Geriatric patients are more likely to have multiple chronic diseases than younger generations; therefore, they need more medications. Taking a higher number of medications is considered to be a risk factor because of more side effects, non-adherence, financial costs, drug–drug interactions, and morbidity outcomes. According to the United Nations, a geriatric is defined as a person 60 years or older, and an adult aged 80 is referred to as an old geriatric [1].

As the world population is aging, with the expectance of people over 65 years old to reach 71 million by 2030, compared with 35 million in 2000; by 2050, the world average life expectancy is predicted to increase by 10 years compared with that in the 2000s. These statistics shows that there will be more medications to be used per person [2,3]. 

Saudi Arabia has a great percentage of geriatric patients associated with multiple chronic diseases who require close attention and close monitoring for their medications. The life expectancy in Saudi Arabia has improved from 64.4 years in the 1980s to 74.3 years in the 2000s (World Bank, 2014) [4]. As a result, the elderly population aged 60 years old and above is projected to increase to 18.4% in 2035 [5].

Polypharmacy is defined as the use of multiple medications by a patient, while others describe polypharmacy as the optimization of medications for a patient to use multiple appropriate medications [6]. The definition of polypharmacy is debatable, as the precise number of multiple medications used to define “polypharmacy” generally ranges from five or more appropriate medications [7,8]. It is very important to focus, “in the definition of polypharmacy”, on prescribed “appropriate” medications, therefore, this study did not consider the number of over-the-counter and herbals/supplements used. 

This study considers “polypharmacy” as a patient receiving five appropriate medications or more. This study also considers geriatrics as subjects 65 years or older. The objective is to target geriatric patients who are suffering from one chronic disease with the possibility of concurrent comorbidity, which is defined as the presence of one or more additional diseases or disorders to the primary chronic disease. Polypharmacy in geriatrics is becoming a global burden because it is associated with the following:Increased healthcare costs for elderly patients, prescribing inappropriate medications will contribute in extra cost for the patients as well as the healthcare system [9].Attributing the risk of adverse drug reaction, the risk of adverse drug reaction (ADR) is increased as the number of medications increases; therefore, it is expected that ADR is higher in geriatric patients [9].Prescription cascade, addition of medications due to misinterpreted ADRs will continue adding medication with being misdiagnosed [7].Drug–drug interaction, elderly patients are predisposed to drug–drug interactions, the probability of drug–drug interactions is increased as the utilization of a number of medications is increased [9].Patient compliance, elderly patients are more commonly not adherent to their medications because of its different and frequent regimens [9].

### Age-Related Pharmacokinetics, Pharmacodynamics

Drug response depends on biological variation as a sequence of events in pharmacogenetics factors, which can play a role as predictors of adverse health outcomes in elderly patients [10]. The main two differences between elderly patients and individual adults are the pharmacokinetics and pharmacodynamics of the medications, where some physiological processes for some drugs tend to change in elderly patients. Pharmacokinetics processes are concluded by decreased absorption, decreased distribution, decreased metabolism, decreased excretion with most drugs, prolonged gastric emptying, and decreased gastric acid secretion [7,11,12].

It has been reported that there is a reduction in liver volume, a loss of volume, and a decline in hepatic blood flow; these factors will negativity affect metabolism of drugs in the liver and increase the incidence of ADR [12]. Moreover, an electrolyte imbalance can decrease the drug clearance rate [13], in addition to poor physical and cognitive capability, which affects dose-dependent relationships [14].

A reduction in renal ability to function should affect clearance of many drugs (specifically water-soluble drugs), prolongation of the drug’s half-life, adverse drug reaction, hospital admission, and most likely toxicity of the drug [15,16], and will induce accumulation of the drugs.

Multi-morbid older people are in a continuous growth in the global population. The issue of polypharmacy is a matter of concern in older people, who tend to have more chronic diseases for which a large number of medications is prescribed. Consequently, it is the ultimate risk factor for inappropriate prescribing, adverse drug reactions and events, and medication underuse and duplication. The relationship between the number of medications and the number of illnesses in correlation with geriatric patients as they age per year is still a major concern.

## 2. Aims & Objectives

The objectives of conducting this overview study are as follows:
To develop a full-framed picture about the utilization of medications for geriatric patientsTo review the percentage of older adults with five medications or more.

The main aim for this study is as follows:To investigate the association between polypharmacy and comorbidities in elderly patients.

## 3. Methodology

### 3.1. Study Sitting and Area

This study was conducted in the Prince Sultan military medical city database during 2018 in Riyadh. More than 4000 medical profiles were obtained to participate in this study, the utilization of medications for geriatric patients. 

### 3.2. Study Design

A retrospective cross-sectional study to evaluate the utilization of medications and comorbidities in geriatric patients.

### 3.3. Study Population 

The geriatric patients’ database conducted among targeted patients aged 65 years or older, longstanding patients who are taking multiple chronic medications for different indications. 

### 3.4. Sample Size 

#### 3.4.1. Population Characteristics

This study was conducted on 3009 patient profiles during 2018. We had a representative sample of (4011 patients) with a dropout of almost 1002 patients profiles that did not meet the study inclusion criteria.

#### 3.4.2. Inclusion Criteria

Male and female patients aged 65 and older.Patients received appropriate medications.

#### 3.4.3. Exclusion Criteria

Patients who are receiving non-systemic or short-duration medications.Male and female patients who are younger than 65 years old.People aged 65 who did not take any medication.Healthy people.

### 3.5. Data Collection Tools and Techniques

#### 3.5.1. Database Collection 

A template for the data collection sheet was structured using Excel to give detailed descriptions of the prescribed medications, in terms of body systems and drug classifications, to acquire complete medication history, along with specifying comorbidities. 

Only qualified and professional candidates were chosen for data entry to present the quality and accuracy of data; aiming to provide comprehensive information about geriatric medications. 

Professional investigators reviewed the geriatric medication records during the past year and described them thoroughly. Then, they selected patients who had five appropriate medications or more and considered them as patients with polypharmacy. 

#### 3.5.2. Medication(s) Count

Investigators defined “appropriate medication” as prescribed medication used in systemic chronic medications with a duration of at least six months, not including herbal/folklore drugs, vitamin/mineral supplements, and other short-duration medications—nor will over the counter drugs be counted as appropriate medications. Other medications such as ophthalmic, topical, intranasal, and other non-systemic medications are not considered as “appropriate medications”. Each appropriate medication being administered to the patient throughout the last year was aggregated and presented as “total” with the exception of “other”. Our classification of the medication was in accordance with the classification of British National Formulary (BNF). 

### 3.6. Statistical Analysis Consideration 

The main variables include the total number and type of medication and comorbidity diseases were collected and saved in an Excel sheet. The variables related to the study analyzed descriptive statistics, frequency, and percentage of medications and comorbidities to show a demographic analysis for the current situation. All information is presented as tables, graphs, and charts where applicable 

### 3.7. Ethical Consideration/Approval 

Ethical clearance certificate to conduct this study was obtained from the Institutional Review Board (IRB) from Princess Nourah University’s health science research center with IRB Log Number: 18-0141. Confidentiality and privacy of the data were assured and declared in the database. The rights, safety, and well-being of the participants, as well as confidentiality of their names, medical record numbers, identity, and identification numbers, are the most important consideration. This study was conducted in compliance with same protocol after the IRB approval.

## 4. Results

This study was conducted on 4011 patient profiles, after which 1002 profiles were disqualified for not meeting exclusion criteria as they did not have appropriate medications. The remaining total was 3009 profiles (having one or more appropriate medication), with 56% males (n = 1685) and 44% females (n = 1324). It was found that 55.7% (n = 1676) of the total patients were receiving more than five proper medications— 53% males and 47% females. The average of patients’ age in years was 73.26 ± 6.6 (SD). There was no significant difference between the mean age of males (73.5 years) and females (72.8 years). The average number of appropriate medications was 5.31 ± 2.8 SD, while the average number of the comorbidities was 2.56 ± 1.25 SD illnesses.

The linear graph (Figure 1) illustrates the average between the number of medications and the average total number of comorbidity diseases. The average number of appropriate medications was diverged to be higher in association with average comorbidities for those patients aged 65–70 years, compared with those patients aged 71 years or more. Polypharmacy was significantly higher in patients aged 65–70 compared with patients aged 71 and older (*p* < 0.001). An average of 6.4 medications was observed for the patients aged 65–70 years compared with an average of 4.2 medications for patients aged 71 years and older; this difference was statistically significant with *p* < 0.01. In the mean time, a linear average of almost 2–3 comorbidity diseases was associated with all elderly patients aged 65 years and older. This linear relationship did not show any significant correlation between age and number of diseases (Figure 1). 

A glance at the bar graph (Figure 2) provided the most existing eight comorbidity diseases among geriatric patients in Saudi Arabia, tracking the whole 3009 profiles patients utilizing medication(s), although it had almost same sequence of utilization with patients with polypharmacy. As a general trend, hypertension was the most common comorbidity disease with more than 47% (1891 patients), followed diabetes mellitus with 37.3% (1496 patients), which almost with same percent as hyperlipidemia with about 36% (1440 patients), considering that most patients had more than one comorbidity disease. Other diseases such as coronary artery disease, thyrosis, benign prostatic hyperplasia, rheumatoid arthritis, and chronic obstructive pulmonary disease were considered less common in elderly patients in Saudi Arabia, as illustrated in Figure 2. Polypharmacy were associated mostly with patients receiving cardiovascular medications and patients receiving endocrine medications as illustrated in Figure 3.

Table 1 shows that elderly patients on cardiovascular drugs (including calcium channel blockers, angiotensin-converting-enzyme inhibitor, angiotensin II receptor blockers, vasodilators, and other cardiovasculars; also including antilipidemics, antithromobotics, and antiarrythmics) had the greatest percentage with 83.9%, where some of these patients received no other drugs. In the mean time, elderly patients on cardiovascular (CV) drugs showed the most patients with polypharmacy. Elderly patients on endocrine medications (including insulin, biguanides, sulfonylureas, and new classes of antidiabetics; as well as antithyroids) were prescribed next to cardiovascular medications with almost 61%, where almost 10% of these patients had polypharmacy. Patients with gastrointestinal (GI) drugs (42.8%), genito-urinary drugs (37%), musculoskeletal drugs (33%), Central Nervous System (CNS) drugs (20.8%), and respiratory drugs (11.6%), where all patients with these comorbidity diseases had less polypharmacy.

Antilipidemics, followed by antithrombotics, were the most extensively used medications in elderly patients. Calcium channel blockers were the most prescribed antihypertensive class, among 43.7% of all hypertensive patients. Correspondingly, biguanides were the most prescribed to the diabetic patients, with nearly 70% percent out of all the other antidiabetic classes.

## 5. Discussion

This study defined the geriatric population as the age group of 65 years and older, where the general authority for statistics in Saudi Arabia 2017 found that the number of those in the elderly population (65 years and over) reached 1,050,885 persons, representing 3.23% of the total population. It showed that the female population was 42.52% and the male population was 57.49%, which reflects the ratio in this study of 44% female and 56% male [17]. Also, polypharmacy is defined as a patient receiving five appropriate medications or more. A somewhat high prevalence of polypharmacy (55%) was shown in the sample. This percentage could be attributed to the fact that the elderly group admitted to hospital was expected to have multiple comorbidities, which could lead to a greater influence in utilization and medication consumption. Because there was a need for a clear and feasible definition of polypharmacy [18,19], the definition from this study concerned only appropriate medications; hence, polypharmacy is common in elderly patients, and nearly 50% of the elderly were taking one or more medications that were not medically necessary. In addition, elderly patients were very keen to purchase over-the-counter medication and herbal products [20].

Data showed that elderly patients aged 65–70 years were associated with more polypharmacy, with an average of 6.4 medications, compared with an average of 4.2 medications for patients aged 71 years and older. The same data showed a persistent comorbidity between two and three diseases for most, if not all elderly patients. The increased risk of undesirable health outcomes and negative consequences associated with polypharmacy such as risk of future adverse health events [21]. 

The general authority of statistics in the Kingdom of Saudi Arabia announced that elderly citizens (65 years and older) showed the highest percentage of chronic diseases in this population restricted to hypertension and diabetes mellitus (DM) (28.5% and 28.7%, respectively), followed by arthritis with 13.9% [17]. While the results of this study on geriatrics’ polypharmacy had similar results, showing that hypertension was the most frequent disease, followed by diabetes mellitus with a similar percentage to hyperlipidemia. It should be kept in mind that some antihypertensive and antihyperlipidemic medications could be dispensed to diabetic patients as a prophylactic, such as angiotensive converting enzyme inhibitors given for DM with normotensive patients.

Morbidity and several chronic diseases have the influence to increase complications in how to manage the disease, specifically in elderly patients. In our study, it was noticed that the number of appropriate medications and number of comorbidity diseases are correlated with age, although there is no correlation between age and diseases, and neither between age and number of medications. These findings were similar to a study that was conducted on elderly patients during their discharge from a tertiary care hospital, which showed that the prevalence of comorbidity in number and type of chronic conditions similar for both men and women and age had no impact on polypharmacy, and also illustrated that the percentage of chronic diseases among patients older than 65 years with hypertension was the greatest, affecting 66.0% of their patients, while the percentage of patients with diabetes was 29.8%, and that with hyperlipidemia was 40.8% [22]; further, antihypertensive agents were the most prescribed medication [23].

Comorbidity, a concept that describes the complication of other disease conditions on geriatric patient, might have another primary condition of interest. The comorbidity–polypharmacy score consists of the sum of all known comorbid conditions and all associated medications. This score is helpful in deciding how to estimate the risk of death and how aggressively to treat a condition [24,25,26]. In this study, we differentiate to find out the correlation between the number of diseases and number of illnesses, and it was obvious and significant difference that those aged 65–70 had a higher number of medications compared with older patients; this difference was statistically significant with *p* < 0.01.

Medications should be prescribed for suitable indications, ensuring that elderly patients are fully aware of the benefits and complications. Electronic-based records for medications give the opportunity to pharmacists and physicians to prescribe, review, verify, and monitor their patients, and allow the identification of the high risk of adverse drug events and complications [27]. This study contraindicated the theory that the “number of medications increased as the patients age increasing” and controverting other studies [22,27]; in the mean time, this study confirmed SIMPATHY (Stimulating Innovation Management of Polypharmacy and Adherence in The Elderly), looking toward the year 2030 to approach and implement medication safety management program [28].

Physicians and pharmacists have the potential to reduce medication errors in elderly patients, reduce number of medications, and reduce adverse events. Simple scenarios could be implemented to eliminate confusion for elderly patients for complex medication regimens or to provide accurate and complete drug instructions and monitoring to patients and their families [29]. It is important to follow the American Society for Hospital Pharmacy recommendations, summarized as not treating symptoms or adverse events, not prescribing more than five medications to a patient, and making prior verification for medication refill [30]. It is the responsibility of pharmacists to educate primary care physicians and elderly patients to ensure the safe, effective, and appropriate use of medications. Pharmacists should target how to improve drug therapy and minimize drug intake in elderly patients with a substantial decrease in the cost of drugs [31].

## 6. Conclusions

Polypharmacy in the elderly population was the concern of this study. Authors provided the percentage (55%) of subjects who were suffering from polypharmacy in the past year. Awareness of geriatric medications and diagnosed diseases will improve in managing ADR and other risk factors. The main investigation of this study was to examine the correlation between polypharmacy and aging in geriatrics. Investigators have provided evidence that polypharmacy is not associated with aging. People aged 71 years and older are not likely to have more health complications than those who aged 65–70 years.

## Figures and Tables

**Figure 1 geriatrics-04-00036-f001:**
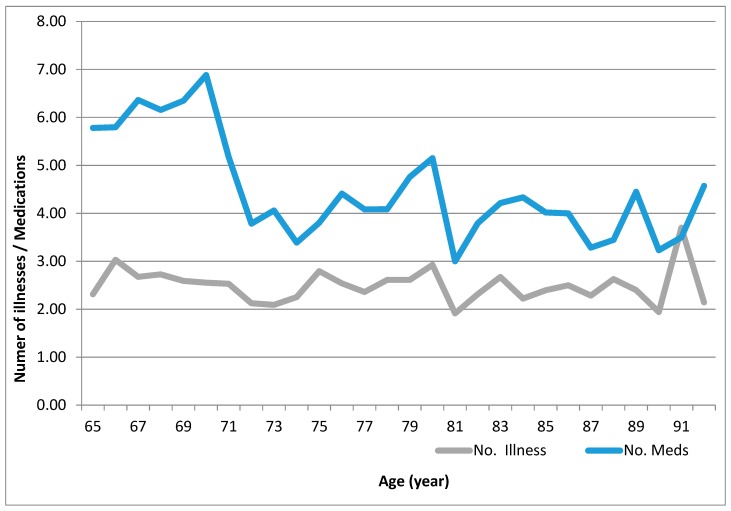
Average number of appropriate medications and number of comorbidity diseases in correlation with age.

**Figure 2 geriatrics-04-00036-f002:**
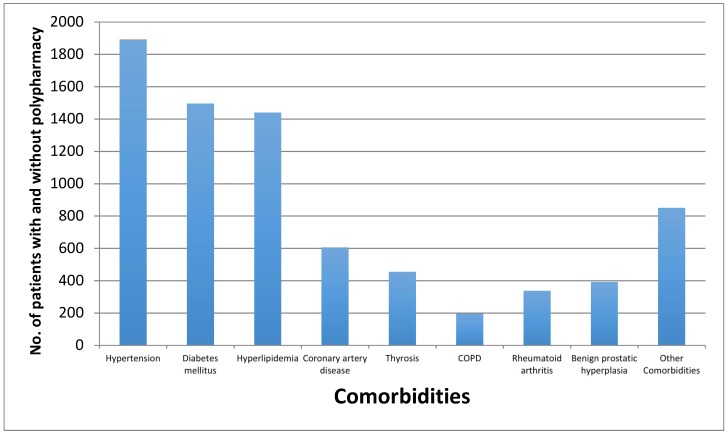
Prevalence of most frequent comorbidity diseases among elderly patients with and without polypharmacy in Saudi Arabia.

**Figure 3 geriatrics-04-00036-f003:**
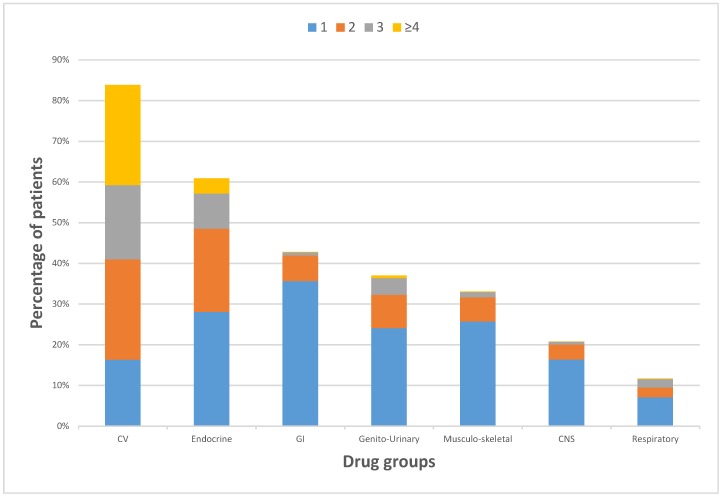
Percentages of most frequently prescribed appropriate medications, from taking 1 to ≥4 drugs per patient (not showing (0 drugs)).

**Table 1 geriatrics-04-00036-t001:** Frequency of patients in each drug quantity taken per patient.

Drugs by Body System: (% of All Patients)	Number of Patients Per Drug Quantity				
	0 Drugs	1 Drug	2 Drugs	3 Drugs	≥4 Drugs
Cardiovascular System Drug(s) (83.88%)	485 patients	491 patients	742 patients	549 patients	742 patients
Endocrine System Drug(s) (60.91%)	1176 patients	845 patients	615 patients	260 patients	113 patients
Gastro-intestinal System Drug(s) (42.83%)	1720 patients	1073 patients	189 patients	24 patients	3 patients
Genito-urinary System Drug(s) (37.02%)	1895 patients	725 patients	247 patients	123 patients	19 patients
Musculoskeletal System Drug(s) (33.06%)	2014 patients	774 patients	178 patients	39 patients	4 patients
Nervous System Drug(s) (20.80%) patients	2383 patients	493 patients	111 patients	20 patients	2 patients
Respiratory System Drug(s) (11.69%)	2657 patients	213 patients	72 patients	63 patients	4 patients

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
