# Peer review of "A Comprehensive Overview of Polypharmacy in Elderly Patients in Saudi Arabia"

_geriatrics, 2019, doi:10.3390/geriatrics4020036_

Round 1

Reviewer 1 Report

 The paper named "A comprehensive overview of polypharmacy in elderly patients in Saudi Arabia" made by Aseel Alsuwaidan et al, is about the prevalence of polypharmacy in elderly population of Saudi Arabia. The topic is interesting but this is a known argument. In the paper is possible only to know about the presence of drugs > 5 and the principle class of drugs (cv, neurologicalecc.).  

MAJOR REVISION:

I suggesto to authors:

- have to use scientifica criteria (for example 2019 "beers criteria" or symilar) to investigate the prevalence of appropriated drugs in the population

- have to use an index of complexity for comorbidities (for example (Charlson index)

- have to use a statistica analysis with significative results (p value) end not only frequence analisys 

MINOR REVISION

- english language, minor spell check required

Author Response

MAJOR REVISION:

I suggesto to authors:

-       have to use scientifica criteria (for example 2019 "beers criteria" or symilar) to investigate the prevalence of appropriated drugs in the population

Done

-       have to use an index of complexity for comorbidities (for example (Charlson index)

Done, used as a reference.

-       have to use a statistica analysis with significative results (p value) end not only frequence analisys 

Done, as appropriate.

MINOR REVISION

-       english language, minor spell check required

Authors don’t mind to pay for editing English language since it is not a mother language for them all.

I would like to thank the reviewer for the valid revision

Reviewer 2 Report

Peer Review Report

Summary:

The study aimed to provide an overview picture about the medication usage in older patients in one of the hospitals in Saudi Arabia. They reported descriptive results about the prevalence of medication usage and its association with age and existing comorbidities. The main findings in this study are that medication usage is prevalent in older patients and that it is not associated with increasing age. Despite the big sample size, I have major concerns about several aspects in this study.

Broad comments:

I have a major concern about the quality of English language throughout the different sections. There are numerous spelling and grammatical mistakes. In addition, the structure of the sentences is incoherent in multiple parts. I suggest the manuscript needs a round of English editing.

The introduction section requires some reformatting and language editing. Moreover, some important references are missing. The flow of the introduction and the ideas the authors want to present are not coherent .The aims paragraph at the end of the introduction is unclear. I have trouble pinpointing the research question that the study was meant to address.

The statistical analysis section is missing essential explanation of the statistical tests used and type of analysis done. The variables used in the analysis need more explanation and clarification. I suggest adding more details about the statistical test used for the analysis, the cut-off values for significant results.

No enough information were provided about the characteristics of the included population in the results section. Because the authors didn’t provide information about their statistical methods, the results they reported are difficult to be interpreted.

More details and information are needed about the characteristics of the study population. I suggest adding a separate table to present this data. Moreover, the results and associations need to be interpreted in a more detailed and clear way. Authors mentioned several associations without providing P values.

Specific comments:

Abstract: line #28, the authors mentioned the prevalence of polypharmacy without mentioning a definition for polypharmacy in the methodology section, I’d suggest adding the definition of polypharmacy to the methods.

Abstract: line #33, was this a finding of the study? There are several definitions for polypharmacy in the literature not just 5 or more medications. I’d suggest summarizing the study findings in the conclusion section.

Introduction: lines 41-44, I’d suggest adding some relevant references here.

Introduction: lines 59-61, can the authors provide a rationale for this sentence as several data have shown that over the counter medications can contribute to adverse events in older adults and could be counted towards polypharmacy:  “In the definition of polypharmacy, it is very important to focus on prescribed 60 “appropriate” medications, so that we should not consider the number of over-the-counter and herbals/supplements used”.

Introduction: line 68, can the authors clarify the following: the authors mentioned here prescribing inappropriate medications will contribute to extra-cost for the patients, this contradicts with the definition the authors provided for polypharmacy (using of appropriate medications).

Introduction: lines 99-104, the organization of the aims was confusing in that it was difficult to determine what the main aim of this study was. The authors mentioned the objectives and the aims in to different ways. The aims section can be clarified in an easier and clearer way.

Methods: line 125, can the authors clarify the following: if the patient is receiving both non systemic and systemic medications, will he/she be included?

Methods: line 128, can the authors provide a definition for “healthy people”?

Methods, line 147, I’d suggest adding a reference here.

Methods: lines 148-152, can the authors clarify the following: the authors mentioned the examined the association between polypharmacy and comorbidities, however no information were provided about the statistical test used for this analysis and also the cutoff value for significant variables. The analysis can be expanded to added further important details

Methods: lines 148-152, can the authors clarify the following: some important information about the used variables are missing regarding how they defined them and if they used continuous or binary variables.

Results: I’d suggest adding a separate table to describe the sample population characteristics.

Results: line #166, Can the authors provide P value for this relation?

Results: line# 173, can the authors clarify the following: the authors mentioned that they found a significant difference between number of medication received by two different age groups with a P value <0.5, I am not sure if this was a typo or the real analysis done.

Author Response

Broad comments:

I have a major concern about the quality of English language throughout the different sections. There are numerous spelling and grammatical mistakes. In addition, the structure of the sentences is incoherent in multiple parts. I suggest the manuscript needs a round of English editing.

The introduction section requires some reformatting and language editing. Moreover, some important references are missing. The flow of the introduction and the ideas the authors want to present are not coherent .The aims paragraph at the end of the introduction is unclear. I have trouble pinpointing the research question that the study was meant to address.

The statistical analysis section is missing essential explanation of the statistical tests used and type of analysis done. The variables used in the analysis need more explanation and clarification. I suggest adding more details about the statistical test used for the analysis, the cut-off values for significant results.

No enough information were provided about the characteristics of the included population in the results section. Because the authors didn’t provide information about their statistical methods, the results they reported are difficult to be interpreted.

More details and information are needed about the characteristics of the study population. I suggest adding a separate table to present this data. Moreover, the results and associations need to be interpreted in a more detailed and clear way. Authors mentioned several associations without providing P values.

 Please find below; perhaps it is the answer for previous comments:

Group of Age

Appropriate Medications 

Total Comorbidities 

Age (in year)

Mean

SD

Mean

SD

65 – 70 (n=1558)

6.35

2.85

2.66

1.25

71 – 75 (n=426)

4.25

2.50

2.43

1.25

76 – 80 (n=475)

4.51

2.48

2.62

1.22

81 – 85 (n=355)

3.81

2.04

2.29

1.30

86 and hi (n=165)

3.62

1.91

2.34

1.12

Total (n=3009)

5.31

2.84

2.56

1.25

I did find this is with benefit for the readers.

Specific comments:

Abstract: line #28, the authors mentioned the prevalence of polypharmacy without mentioning a definition for polypharmacy in the methodology section, I’d suggest adding the definition of polypharmacy to the methods.

The definition was mentioned in the conclusion because this is the first time used, that is why it is part of the conclusion. This definition expressed ONLY appropriate medications (NOT ONLY prescribed so that this study excluded the OTC.

Abstract: line #33, was this a finding of the study? There are several definitions for polypharmacy in the literature not just 5 or more medications. I’d suggest summarizing the study findings in the conclusion section.

Pharmacists, who can differentiate the appropriate medications, entered the data for this study.  

Introduction: lines 41-44, I’d suggest adding some relevant references here.

The reviewer is absolutely right, yet this is general statement that authors used it as introduction; more details for this statement 67-80, and part 1.1.

Introduction: lines 59-61, can the authors provide a rationale for this sentence as several data have shown that over the counter medications can contribute to adverse events in older adults and could be counted towards polypharmacy:  “In the definition of polypharmacy, it is very important to focus on prescribed 60 “appropriate” medications, so that we should not consider the number of over-the-counter and herbals/supplements used”.

The reviewer is right, specifically if we are dealing with elderly patients that OTC meds could contribute to adverse affect or even in drug-drug interaction.  The authors cannot control the OTC neither herbal medicine (which is most often used in elderly patients in KSA).  This is the justification for why did not consider this study OTC neither herbal medicine in the data?

Introduction: line 68, can the authors clarify the following: the authors mentioned here prescribing inappropriate medications will contribute to extra-cost for the patients, this contradicts with the definition the authors provided for polypharmacy (using of appropriate medications).

Noted and changed to be:prescribing more medications will contribute in extra cost for the patients …..

Introduction: lines 99-104, the organization of the aims was confusing in that it was difficult to determine what the main aim of this study was. The authors mentioned the objectives and the aims in to different ways. The aims section can be clarified in an easier and clearer way.

Noted and changed to be:The main aim for this study is to have full-framed picture about the utilization of medications for geriatric patients and how to improve health care management in geriatrics. 

Methods: line 125, can the authors clarify the following: if the patient is receiving both non systemic and systemic medications, will he/she be included?

Authors clarified that in exclusion criteria

Methods: line 128, can the authors provide a definition for “healthy people”?

Noted, it is the same as the prior line, therefore it is deleted.

Methods, line 147, I’d suggest adding a reference here.

Noted

Methods: lines 148-152, can the authors clarify the following: the authors mentioned the examined the association between polypharmacy and comorbidities, however no information were provided about the statistical test used for this analysis and also the cutoff value for significant variables. The analysis can be expanded to added further important details

It has been described in the discussion part.

Methods: lines 148-152, can the authors clarify the following: some important information about the used variables are missing regarding how they defined them and if they used continuous or binary variables.

Results: I’d suggest adding a separate table to describe the sample population characteristics.

Please see the above table.

Results: line #166, Can the authors provide P value for this relation?

Noted and changed.

Results: line# 173, can the authors clarify the following: the authors mentioned that they found a significant difference between number of medication received by two different age groups with a P value <0.5, I am not sure if this was a typo or the real analysis done.

Noted, I assure the reviewer that I went back for the results and make further analysis to verify the results. 

Round 2

Reviewer 1 Report

I apprecciated changing effectuated. I think the paper can be accepted in this last form,

Reviewer 2 Report

I think the authors did a good job in terms of addressing the comments and making required edits especially the English language changes.